# OVIR-3D: Open-Vocabulary 3D Instance Retrieval Without Training on 3D Data

**Shiyang Lu   Haonan Chang   Eric Pu Jing   Abdeslam Boularias   Kostas Bekris**
Rutgers University
https://github.com/shiyoung77/OVIR-3D

**Abstract:** This work presents OVIR-3D, a straightforward yet effective method for open-vocabulary 3D object instance retrieval without using any 3D data for training. Given a language query, the proposed method is able to return a ranked set of 3D object instance segments based on the feature similarity of the instance and the text query. This is achieved by a multi-view fusion of text-aligned 2D region proposals into 3D space, where the 2D region proposal network could leverage 2D datasets, which are more accessible and typically larger than 3D datasets. The proposed fusion process is efficient as it can be performed in real-time for most indoor 3D scenes and does not require additional training in 3D space. Experiments on public datasets and a real robot show the effectiveness of the method and its potential for applications in robot navigation and manipulation.

**Keywords:** Open Vocabulary, 3D Instance Retrieval

## 1   Introduction

There has been recent progress in open-vocabulary 2D detection and segmentation methods [1, 2, 3] that rely on pre-trained vision-language models [4, 5, 6]. However, their counterparts in the 3D domain have not been extensively explored. One reason is the lack of large 3D datasets with sufficient object diversity for training open-vocabulary models. Early approaches for dense semantic mapping [7, 8, 9, 10] project multi-view 2D detections to 3D using closed-set detectors but cannot handle arbitrary language queries. More recently, OpenScene [11] and Clip-fields [12] achieve open-vocabulary 3D semantic segmentation by projecting text-aligned pixel features to 3D points and distilling 3D features from the aggregated 2D features. Given a text query during inference, OpenScene [11] generates a heatmap of the point cloud based on the similarity between point features and the query feature. Nevertheless, manual thresholding is required for object search to convert a heatmap to a binary mask and it lacks the ability to separate instances from the same category. This limits its use in robotic applications, such as autonomous robotic manipulation and navigation. Clip-fields [12], on the other hand, requires training on additional ground truth annotation for instance identification, which makes it less open-vocabulary at the instance level.

This work focuses on open-vocabulary 3D instance retrieval, aiming to return a ranked set of 3D instance segments given a 3D point cloud reconstructed from an RGB-D video and a language query. Some examples are shown in Figure 1. While 2D segmentation from a single viewpoint is often insufficient for robot grasping and navigation, 3D instance retrieval methods generate a more complete and accurate segmentation of objects in 3D space by multi-view fusion and smoothing. In particular, this work considers a scenario, where a mobile robot navigates in an indoor scene and automatically reconstructs the 3D environment using its RGB-D sensor and an off-the-shelf SLAM module. Instead of fusing pixel-level information and then either grouping them into instances by thresholding at inference time [11] or training an object identification model with additional ground truth data [12], the proposed method directly fuses instance-level information into the 3D scene without additional training, so that given a text query such as "lamp" or "bed", a robot can immediately locate the top-related object instances and perform required tasks.

7th Conference on Robot Learning (CoRL 2023), Atlanta, USA.

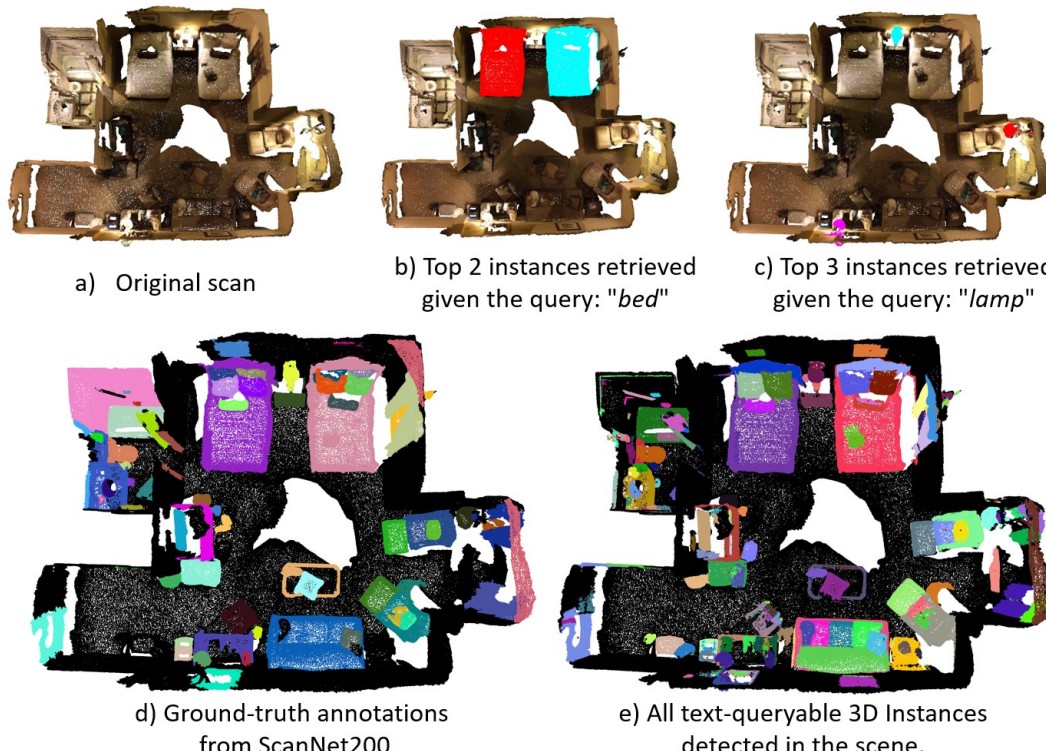

a) Original scan

b) Top 2 instances retrieved given the query: "*bed*"

c) Top 3 instances retrieved given the query: "*lamp*"

d) Ground-truth annotations from ScanNet200

e) All text-queryable 3D Instances detected in the scene.

Figure 1: **Examples of open-vocabulary 3D instance retrieval using the proposed system.** (a-c) Given a 3D scan reconstructed from an RGB-D video (e.g., scene0645 from ScanNet [13]) and a text query (e.g., bed, lamp), the proposed method retrieves a set of 3D instances ranked based on their semantic similarity to the text query. (d-e) Instances that are not even in the ground-truth annotations can also be detected and queried by the proposed method, such as the cushions on the sofa.

The proposed method addresses this problem by first generating 2D object region proposals and their corresponding text-aligned features by querying a 2D open-vocabulary detector with an extensive vocabulary. It then performs data association and periodic filtering and merging of 3D instances to improve instance masks and remove noisy detections. Finally, a post-processing step handles isolated objects and filters small segments that are likely to be noise. Extensive experiments on real scans from both room-scale dataset ScanNet200 [14] and tabletop-scale dataset YCB-Video [15] demonstrate the effectiveness of the proposed method, which offers an efficient 2D-to-3D instance fusion module ($\sim$ 30 fps for a scene in ScanNet [13] on an NVIDIA RTX 3090) and an open-vocabulary 3D instance retrieval method with near-instant inference time for a text query.

The main contributions of this work are: (i) an efficient 2D-to-3D instance fusion module given text-aligned region proposals, which results in (ii) an open-vocabulary 3D instance retrieval method that ranks 3D instances based on semantic similarity given a text query.

## 2 Related Work

### 2.1 2D Open-Vocabulary Detection and Segmentation

With the advent of large vision-language pre-trained models, such as CLIP [4], ALIGN [5] and LiT [16], a number of 2D open-vocabulary object detection and segmentation methods have been proposed [17, 18, 19, 1, 2, 20, 21]. For 2D semantic segmentation, LSeg [17] encodes 2D images and aligns pixel features with segment label embeddings. OpenSeg [18] uses image-level supervision, such as caption text, which allows scaling up training data. GroupViT [19] performs bottom-up hierarchical spatial grouping of semantically-related visual regions for semantic segmentation. For 2D object detection, ViLD [1] achieves open-vocabulary detection by aligning the features of class-agnostic region proposals with text label features. Detic [2] attempts to address the long-tail detection problem by utilizing data with bounding box annotations and image-level annotations.

OWL-ViT [20] proposes a pipeline for transferring image-text models to open-vocabulary object detection. Our proposed method adopts Detic [2] as a backbone detector to locate objects in 2D images since it can provide pixel-level instance segmentation and text-aligned features. Furthermore, it can be queried with a large vocabulary without sacrificing much speed.

## 2.2 3D Reconstruction and Closed-Vocabulary Semantic Mapping

Early works have addressed the 3D reconstruction problem either through online SLAM methods [22, 23, 24, 25, 26] or offline methods like structure-from-motion [27, 28] using a variety of 3D representations, such as TSDF [29], Surfel [30], and more recently NeRF [31, 32, 33, 34]. With the advancement of learning-based 2D object detection and segmentation methods, recent efforts have focused on point-wise dense semantic mapping of 3D scenes [7, 8, 10, 9, 35]. Despite being effective, these methods have not yet been designed to fit open-vocabulary detectors. They either assume mutually exclusive instances [7, 10] or utilize category labels for data association [8, 9, 35]. The proposed method in this paper adopts an off-the-shelf 3D reconstruction method and focuses on integrating 2D information with point-cloud information to achieve open-vocabulary 3D instance segmentation. A key contribution in this context is a method that associates open-vocabulary 2D instance detections and fuses them into a 3D point cloud while keeping them open-vocabulary.

## 2.3 3D Open-Vocabulary Scene Understanding

More recently, research efforts aim for open-vocabulary 3D scene understanding [12, 11, 36, 37, 38, 39, 40]. Given that existing 3D datasets tend to be significantly smaller than 2D image datasets, this is mainly accomplished by fusing pretrained 2D image features into 3D reconstructions. OpenScene [11] projects pixel-wise features from 2D open-vocabulary segmentation models [17, 18] to a 3D reconstruction and distills 3D features for better semantic segmentation. ConceptFusion [38] fuses multi-modal features, such as sound, from off-the-shelf foundation models that can only produce image-level embeddings. LeRF [39] fuses multi-scale CLIP features to a neural radiance field for open-vocabulary query. These methods can generate a heatmap of a scene that corresponds to a query, but they do not provide instance-level segmentation, which limits their use in tasks that require a robot to interact with specific object instances. PLA [40] constructs hierarchical 3D-text pairs for 3D open-world learning and aims to perform not only 3D semantic segmentation but instance segmentation as well. Nevertheless, the method so far has been demonstrated only on certain furniture-scale objects, and performance in other categories is unclear. On the other hand, our method focuses on instance-level, open-vocabulary 3D segmentation without manual 3D annotation.

## 3 Problem Formulation

A 3D scan $\mathcal{X}^N$ represented by $N$ points is reconstructed from an RGB-D video $\mathcal{V} = \{\mathcal{I}_1, \mathcal{I}_2, \ldots, \mathcal{I}_T\}$ given known camera intrinsics $C$ and camera poses $P_t$, where $\mathcal{I}_t$ is the video frame at time $t$. The objective in open-vocabulary 3D instance retrieval is to return a set of $K$ ranked instances represented as binary 3D masks $\mathcal{M}^N = \{m_i | i \in [1, K]\}$ over the 3D scan $\mathcal{X}^N$, given a text query $Q$ and the desired number of instances $K$ to be retrieved. The ranking of instance masks is based on the semantic similarity between the 3D instance and the text query, where the most similar instance should be ranked first.

## 4 Method

The overall pipeline of the proposed method is illustrated in figure 2. To summarize, given a video frame, the method first generates 2D region proposals $\mathcal{R}^{2D} = \{r_1, .., r_k\}$ with text-aligned features $F^{2D} = \{f_1^{2D}, .., f_k^{2D}\}$ using an off-the-shelf 2D open-vocabulary method trained on large 2D datasets. The 2D region proposals $\mathcal{R}^{2D}$ of each frame $\mathcal{I}_t$ are then projected to the reconstructed 3D point cloud given the camera intrinsics $C$ and poses $P_t$. The projected 3D regions $\mathcal{R}^{3D}$ are either matched to existing 3D object instances $O = \{o_1, .., o_b\}$ with 3D features $F^{3D} = \{f_1^{3D}, .., f_b^{3D}\}$ stored in the memory bank $\mathcal{B}$, or added as a new instance if not matched with anything. The 2D region to 3D instance matching is based on feature similarity $s_{ij} = cos(f_i^{2D}, f_j^{3D})$ and region overlapping $IoU(r_i^{3D}, o_j)$ in the 3D space. Matched regions are integrated into the 3D instance. To remove unreliable detections and improve segmentation quality, periodic filtering and merging of

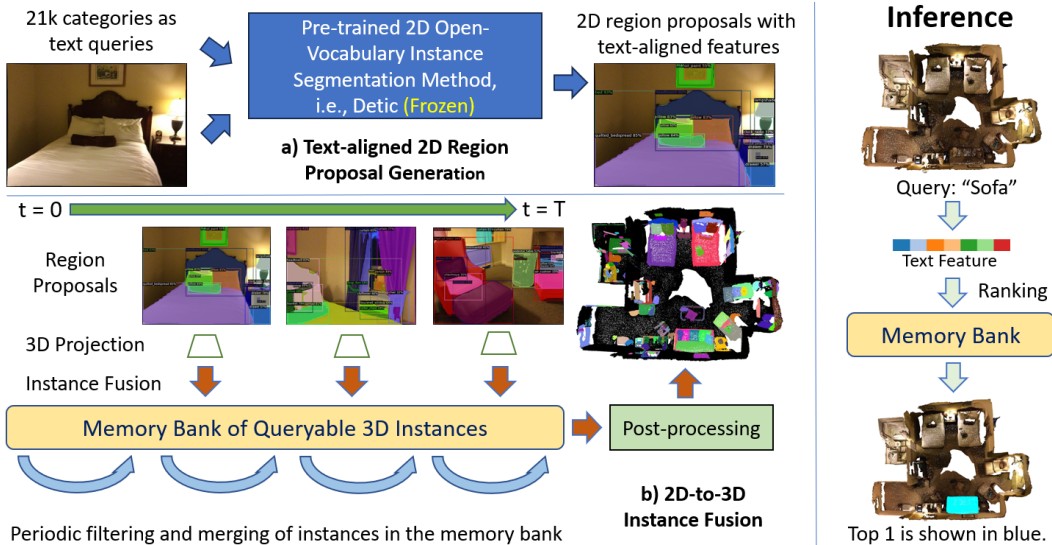

Figure 2: **Pipeline of the proposed method.**

3D instances in the memory bank $\mathcal{B}$ is performed every $T$ frames. A final post-processing step removes 3D instances that are too small and separates object instances that are isolated in 3D space but incorrectly merged. During inference time, the text query $q$ will be used to match with a set of representative features of each 3D instance, and the instances $O$ will be ranked based on the similarity and returned. Details of the proposed method are presented below.

### 4.1 Text-aligned 2D Region Proposal

Learning-based region proposal networks have served as a critical module for many instance segmentation methods, such as MaskRCNN [41]. However, directly generating 3D region proposals for open-vocabulary instance retrieval is hard due to the lack of annotated 3D data with enough category varieties. This work views the 3D region proposal problem as a fusion problem from 2D region proposals. In particular, it leverages the power of an off-the-shelf open-vocabulary 2D detector Detic [2], which is trained with multiple large image datasets, to generate 2D region proposals (masks) $\mathcal{R}^{2D}$ by querying it with an extensive number of categories, i.e. all 21k categories from ImageNet21k [42] dataset. In addition to 2D masks $\mathcal{R}^{2D}$, associated text-aligned features $F^{2D}$ are extracted before the final classification layer of the 2D model. The output category labels are dropped without being used as they are rather noisy given the input vocabulary size. Both region proposals $\mathcal{R}^{2D}$ and their text-aligned features $F^{2D}$ are used for data association in the fusion step.

Though the proposed method does not critically depend on a specific 2D detector, not all open-vocabulary 2D detectors can serve as a region proposal backbone. There are two requirements that a 2D detector must exhibit: 1) It must be able to generate pixel-wise masks for a wide variety of objects in a timely manner; and 2) It should provide a text-aligned feature for each region so that it can be queried with language. During the development of this work, it's found that Detic [2] naturally meets the requirement without additional modifications and thus it is adopted for this work. Some other options, such as SAM [43] and Grounding-DINO [44] were not adopted because they are either too slow or unable to directly output text-aligned features for proposed regions, which turn out to be critical for data association in the experiments. Detic [2] also has a fast inference speed even when queried with all the categories from ImageNet21k [42] ($\sim$ 10fps on an NVIDIA RTX3090), which makes it favorable for this task.

### 4.2 2D-to-3D Instance Fusion

2D region proposals $R^{2D} = \{r_1^{2D}, .., r_k^{2D}\}$ and their corresponding features $F^{2D} = \{f_1^{2D}, .., f_k^{2D}\}$ for each frame $\mathcal{I}_t$ are first projected to the 3D scan using camera intrinsics $C$ and pose $P_t$. The projected 3D regions $\mathcal{R}^{3D}$ are either matched to existing 3D object instances $O = \{o_1, .., o_b\}$ with 3D features $F^{3D} = \{f_1^{3D}, .., f_b^{3D}\}$, where $b$ is the number of 3D instances already stored in the

memory bank $\mathcal{B}$, or added as a new instance if it is not matched with anything. The 3D feature $f_i^{3D}$ of instance $i$ for matching is simply the average of 2D features $f_j^{2D}$ from all the associated 2D regions, i.e. $f_i^{3D} = \frac{1}{n}\sum_{j=1}^n f_j^{2D}$, where $j$ is the index of 2D regions that are associated to 3D instance $i$, and $n$ is the number of 2D regions that have already been merged. The memory bank is empty at the beginning.

The matching of 2D region $r_i$ to 3D instance $o_j$ is based on cosine similarity $s_{ij} = cos(f_i^{2D}, f_j^{3D})$ and 3D intersection over union $IoU(r_i^{3D}, \hat{o}_j)$ between the projected region $r_i^{3D}$ and visible part of the 3D instance $\hat{o}_j$ in the current frame. If $s_{ij}$ is greater than a predefined threshold $\theta_s$ (default $\theta_s = 0.75$) and the overlapping $IoU(r_i^{3D}, \hat{o}_j)$ is also greater than predefined threshold $\theta_{iou}$ (default $\theta_{iou} = 0.25$), then they are considered as a match. Matched regions will be aggregated to the 3D instance, i.e., $o_j := o_j \cup r_i^{3D}$ and $f_j^{3D} := \frac{n}{n+1}f_j^{3D} + \frac{1}{n}f_i^{2D}$. The matching is not restricted to one-to-one as multiple 2D region proposals may correspond to the same instance.

### 4.3  Periodic 3D Instance Filtering and Merging

The fusion process is fast but it will generate redundant 3D instances when a 2D region proposal fails to match properly, potentially leading to low-quality segmentation and inaccurate data association. To address this, periodic filtering and merging of 3D instances stored in memory bank $\mathcal{B}$ occur every $T$ (default $T = 300$) frames. Point filtering is based on the detection rate $r_p^{det}$ of a point $p$, where $r_p^{det} = c_p^{o_i}/c_p^{vis}$. Intuitively, this is the frequency of a point being considered as part of the instance ($c_p^{o_i}$) over the frequency of it being visible ($c_p^{vis}$). Points with $r_p^{det} < \theta_{det}$ (default $\theta_{det} = 0.2$) are removed from instance $o_i$. Meanwhile, the number of points in each projected 3D segment from a single view that corresponds to $o_i$ is recorded. If after point filtering, instance $o_i$ contains fewer than the median number of points in its corresponding segments, then it is filtered entirely. This dynamic threshold that automatically adapts to instance sizes is critical in the filtering process.

Merging of two instances $o_p, o_q$ is determined by feature similarity $s_{pq} = cos(f_p^{3D}, f_q^{3D})$ and 3D intersection over union $IoU(o_p, o_p)$ between two instance segments, using the same thresholds $\theta_s$ and $\theta_{iou}$ as in Section 4.2. Additionally, instances $o_p$ and $o_q$ are merged if $recall(o_p, o_q) = |o_p \cup o_q|/|o_q| \geq \theta_{recall}$ (default $\theta_{recall} = 0.25$) and $s_{pq} \geq \theta_s$, indicating that $o_q$ is mostly contained in $o_p$ and both instances have similar features.

Hyper-parameters are justified through ablation studies in Section 6, and these values remain fixed for experiments in Section 5 across different datasets.

### 4.4  Post-processing

A simple post-processing step is executed to separate object instances that are isolated in 3D space and filter small segments that are likely to be noise. This is achieved by using DBSCAN [45] to find 3D point clusters in each instance, where the distance parameter $eps$ is set to $10cm$. If an instance $o_i$ has segments not connected in 3D space, DBSCAN will return more than one point cluster and $o_i$ will be separated in multiple instances.

### 4.5  Inference

During inference time, a text query $q$ is converted to a feature vector $f_q = \Theta(q)$ using CLIP [4]. Instead of representing each 3D instance with the average feature of associated 2D regions, the $K$ clustering centers by K-Means of associated features, which can be viewed as representative features from a set of viewpoints, are used. The 3D instances are then ranked by the largest cosine similarity $s$ between the text query $q$ and $K$ representative features of an instance. An ablation study on different strategies of feature ensemble is presented in section 6.3.

## 5  Experiments

### 5.1  Datasets

The first dataset used for the experiment is ScanNet200 [14], which contains a validation set of 312 indoor scans with 200 categories of objects. Uncountable categories "floor", "wall", and "ceiling" and their subcategories are not evaluated. The second dataset is YCB-Video [15], which contains a validation set of 12 videos. It's a tabletop dataset that was originally designed for object 6DoF

pose estimation for robot manipulation. The 3D scans of the tabletop scene are reconstructed by KinectFusion [22]. The ground truth instance segmentation labels are automatically generated given the object mesh models and annotated 6DoF poses.

## 5.2 Metrics

Standard mean average precision ($mAP$) metric for instance retrieval at different IoU thresholds is adopted for the evaluation purpose. In particular, $mAP_{25}$ and $mAP_{50}$ at the IoU threshold $\theta = 0.25$ and $\theta = 0.5$ respectively, as well as the overall $mAP$, i.e $\frac{1}{10}\sum mAP_\theta$, where $\theta = [0.5:0.05:0.95]$ are reported. Only annotated object categories in a 3D scene are used as text queries for evaluation. The mAP results were computed for each 3D scene and then averaged for the whole dataset.

## 5.3 Baselines

OpenScene [11], which is the most relevant work to date, is used as the first comparison point. Given an object query, it returns a heatmap of the input point cloud. A set of thresholds $\theta = [0.5:0.03:0.9]$ are tested for each category to convert the heatmap into a binary mask and then foreground points are clustered into 3D instances using DBSCAN [45], similar to the post-processing step in section 4.4. The one with the best overall performance is reported. Furthermore, a series of prior research has focused on semantic mapping using closed-vocabulary detectors. Two representative works, Fusion++ [9] and PanopticFusion [10], are used as comparison points with two revisions: 1) Instead of using their whole SLAM system, this work assumes the 3D reconstruction and ground truth camera poses are given, and only tested their data association and instance mapping algorithms. 2) Their backbone detector MaskRCNN [41] is replaced with Detic [2] for open-vocabulary detection, and the mean feature of associated 2D detections for each instance is used to match text queries.

| | ScanNet200 [14] | | | YCB-Video [15] | | |
|---|---|---|---|---|---|---|
| **Method** | $mAP_{25}$ | $mAP_{50}$ | $mAP$ | $mAP_{25}$ | $mAP_{50}$ | $mAP$ |
| OpenScene [11] | 0.268 | 0.190 | 0.089 | 0.421 | 0.333 | 0.116 |
| *Fusion++ [9] | 0.414 | 0.253 | 0.094 | 0.817 | 0.464 | 0.120 |
| *PanopticFusion [10] | 0.539 | 0.370 | 0.150 | 0.851 | 0.803 | 0.393 |
| **Ours** | **0.564** | **0.443** | **0.211** | **0.863** | **0.848** | **0.465** |

Table 1: Results on ScanNet200 [14] and YCB-Video [15] dataset

| | $mAP_{50}$ | | | $mAP$ | | |
|---|---|---|---|---|---|---|
| **Method** | head | common | tail | head | common | tail |
| OpenScene [11] | 0.308 | 0.178 | 0.067 | 0.150 | 0.076 | 0.033 |
| *Fusion++ [9] | 0.235 | 0.243 | 0.288 | 0.094 | 0.090 | 0.098 |
| *PanopticFusion [10] | 0.335 | 0.360 | 0.424 | 0.145 | 0.146 | 0.162 |
| **Ours** | **0.417** | **0.433** | **0.469** | **0.224** | **0.214** | **0.193** |

Table 2: Results on three sets of categories with different frequencies in ScanNet200 [14]

## 5.4 Results

Quantitative results of instance retrieval on ScanNet200 [14] and YCB-video [15] datasets are shown in Table 1. Furthermore, results on different sets of categories with different frequencies in ScanNet200 are shown in Table 2. The proposed method outperforms all other baselines by a large margin in terms of instance retrieval $mAP$. It seems that OpenScene[11] does not perform well on this task even with an automatically tuned threshold for each category because fused point features are not distinguishable enough. As a result, grouping points into segments with accurate boundaries by thresholding is rather difficult. The proposed method, on the other hand, directly fuses instance-level information and improves segment quality by periodic merging and filtering. The proposed method outperforms the other two baselines primarily because of the use of instance feature similarity as an additional metric for data association while the baselines only consider 3D overlapping, which can easily fail when the 2D detections are noisy, especially in the open-vocabulary setup.

## 5.5 Running Time

The inference time is nearly instant ($\sim 20ms$) for a text query. The running time for the fusion process depends on the number of detections ($N$) in a frame and the number of fused 3D instances ($M$), i.e. $O(MN)$. As mentioned in Section 4.2, it requires two dot products to compute the region overlapping and feature similarity, which in practice is a fast process that operates at $\sim 30FPS$ with an NVIDIA RTX 3090 GPU for most 3D scans in the ScanNet200 [14] dataset.

# 6 Ablation Studies

## 6.1 Input queries to the 2D region proposal method

The proposed method utilizes an open-vocabulary 2D detector as a region proposal method by querying it with a large vocabulary. One concern is whether the input query of the 2D method would affect the performance of the 3D instance retrieval. This ablation study tests queries from multiple datasets as input to the region proposal method and displays their impact on the overall performance. In addition to ScanNet200 [14] and ImageNet21K [42], COCO [46] (80 categories), LVIS [47] (1203 categories), and more aggressively, queries with ImageNet21k categories but without ScanNet200 categories are tested. Results of 3D instance retrieval on the ScanNet200 dataset are shown in Table 4. It turns out that an extensive vocabulary is helpful for generating regions of interest for arbitrary objects. Furthermore, the results show that the region proposal network has certain generalizability, such that even when ScanNet200 categories are completely removed from the ImageNet21k categories, it can still find most regions based on similar categories in the vocabulary, and the final performance of retrieving objects in ScanNet200 only slightly dropped.

|  | COCO | ScanNet200 | LVIS | ImageNet21k | ImageNet21k - ScanNet200 |
|---|---|---|---|---|---|
| $mAP_{50}$ | 0.228 | 0.419 | 0.429 | **0.443** | 0.410 |

Table 3: Results on ScanNet200 [14] with different input queries to the 2D region proposal network

## 6.2 Instance features and 2D masks

In this ablation study, alternative approaches are explored to replace instance features and 2D masks derived from Detic in order to demonstrate the adaptability of the proposed method across different backbones. Rather than utilizing the mask feature directly extracted from Detic, detected bounding boxes are cropped and fed to the CLIP model to extract features from these cropped regions. For the generation of 2D masks, SAM [43] is adopted to create segmentations for the regions proposed by Detic bounding boxes. In this experimental setup, instance fusion is performed every three frames, primarily due to the relatively slow performance of SAM. Substituting the Detic feature with the CLIP feature from cropped images yields slightly inferior results, whereas replacing the Detic mask with the SAM mask leads to an improvement in performance. This outcome is expected since SAM generally produces higher-quality masks, albeit with a tradeoff in terms of speed. It is anticipated that as open-vocabulary 2D detection techniques advance, more potent and efficient methods may emerge as viable alternatives to the current backbone.

|  | Proposed | w/ CLIP feature | w/ SAM segmentation |
|---|---|---|---|
| $mAP_{50}$ | 0.414 | 0.406 | **0.440** |

Table 4: Results on ScanNet200 [14] with different instance features and 2D masks

## 6.3 Feature ensemble strategies

Three different feature ensemble strategies are tested to represent a 3D instance based on associated 2D features. The first strategy is to compute the average of all 2D features. The second strategy involves clustering the 2D features from different viewpoints using the K-Means algorithm, and the clustering centers are used to represent each instance. During instance retrieval, the feature similarity is determined as the maximum similarity between the query feature and the clustering centers. The third strategy is to use the feature from the largest associated 2D region. Results of 3D instance retrieval on the ScanNet200 dataset are presented in Table 5. The approach of using multiple features through clustering outperforms simple averaging, while using the feature from the largest associated 2D region yields the poorest results.

|  | average | clustered (K=16) | clustered (K=64) | feature from largest 2D detection |
|---|---|---|---|---|
| $mAP_{50}$ | 0.428 | 0.429 | **0.443** | 0.380 |

Table 5: Results on ScanNet200 [14] with different feature ensemble strategies

## 6.4 Time intervals and visibility threshold for periodic instance filtering and merging

This ablation study tested different time intervals $T$ and visibility threshold $\theta_{vis}$ for filtering mentioned in section 4.3. Results of 3D instance retrieval on the ScanNet200 dataset are shown in

Table 6 and Table 7 respectively. The frame interval $T = 300$ and visibility threshold $\theta_{vis} = 0.2$ yields the best results.

| | $T = 1$ | $T = 100$ | $T = 300$ | $T = 500$ | $T = 1000$ |
|---|---|---|---|---|---|
| $mAP_{50}$ | 0.340 | 0.417 | **0.443** | 0.410 | 0.412 |

Table 6: Results on ScanNet200 [14] with different time intervals of periodic filtering and merging

| | $\theta_{vis} = 0$ | $\theta_{vis} = 0.1$ | $\theta_{vis} = 0.15$ | $\theta_{vis} = 0.2$ | $\theta_{vis} = 0.25$ | $\theta_{vis} = 0.3$ |
|---|---|---|---|---|---|---|
| $mAP_{50}$ | 0.256 | 0.386 | 0.407 | **0.443** | 0.418 | 0.408 |

Table 7: $mAP_{50}$ on ScanNet200 [14] dataset

# 7 Robotics Experiments

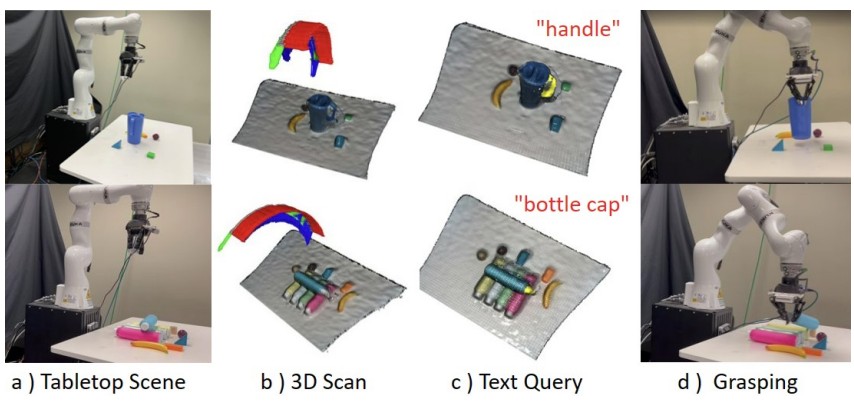

a ) Tabletop Scene     b ) 3D Scan     c ) Text Query     d ) Grasping

Figure 3: **Robot Experiments.**

Contrasting to conventional closed-set semantic methods, the superiority of open-vocabulary detectors for manipulation lies in their ability to pinpoint the grasp region with a language specification. A part-based grasping experiment was devised given this inspiration. In particular, the robot is asked to grasp the "bottle cap" and "handle" respectively in two sets of experiments with five distinct table setups in each set. An RGB-D camera is mounted on the robot's wrist that captures videos of the objects on the table. For each scene, there is a short scanning phase as shown in Figure 3(b) that the robot arm went through a predefined trajectory to get a more complete view of objects on the table. Reconstruction is performed using KinectFusion. OVIR-3D is compared against OpenScene to segment parts given the text query and located segments are used to guide the robot's grasp. For OVIR-3D, the 5/5 graspable bottle caps and 4/5 "handle" of the pitcher were detected and the robot grasping success rate was 90%. The detection rate for OpenScene was low at 0/5 and 3/5 respectively, and the overall grasping success rate was 30%. An object is considered detected if a reasonable visual segment is found.

# 8 Conclusion and Limitations

This paper presents OVIR-3D, a rather straightforward but effective method for open-vocabulary 3D instance retrieval. By utilizing an off-the-shelf open-vocabulary 2D instance segmentation method for region proposal and fusing its output 2D regions and text-aligned features in 3D space, the proposed method can achieve much better performance than other baselines without using any 3D instance annotation, additional training, or manual heatmap thresholding during inference. This method can also be used for 3D instance pseudo-label generation for self-supervised learning.

A limitation of the proposed method is that it is not always able to merge segments of very large instances, such as long dining tables. It can also miss tiny objects as they are likely to be treated as noise and removed during the fusion process. Furthermore, while the proposed method can improve segmentation quality due to multi-view noisy filtering, it still relies on the 2D region proposal model to not consistently miss an object or generate bad segmentation, since it does not use any 3D data for fine-tuning. A promising direction is to integrate this method with a 3D learning-based method to utilize the scarcer but cleaner 3D annotations.

**Acknowledgments**

We thank Dr. Yu Wu for providing useful suggestions during the development of this work. This work is supported by NSF awards 2309866, 1846043, and 2132972.

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
