# OpenReview forum: "OVIR-3D: Open-Vocabulary 3D Instance Retrieval Without Training on 3D Data"
_robot-learning.org/CoRL/2023/Conference — CoRL 2023 Poster_

### Official Review · Reviewer_Qx8o · 2023-07-11

**Confidence:** 4
**Originality:** Good
**Technical Quality:** Very Good
**Clarity Of Presentation:** Very Good
**Impact:** 3

**Recommendation:**

Weak Accept: I recommend accepting the paper, but will not argue for my recommendation if the majority of other reviewers have a different opinion.

**Review:**

Strengths:

1) Open vocabulary instance segmentation (as pioneered in this paper) is a powerful tool that will give robots the ability to find and manipulate arbitrary objects present in a scene. This paper highlights the need for instance segmentation for this purpose (as opposed to the heatmaps produced by previous work).

2) The method is technically sound and is backed up by adequate experimental results.

3) Real world robot experiments. The demonstration of the ability to identify grasp regions of arbitrary objects with language specifications will likely have high impact.

4) The paper is well written, well organized and was a pleasure to read.


Weaknesses:

1) The authors complain of a related work [14] needing a single manual threshold to generate it's output (line 39,40), presenting it as a weakness. However, the method proposed in this paper consists of many manual thresholds: at least 10 (lines 171, 172, 178, 195, 200, 203, 206, 207, 213, 215). It is likely that all these thresholds will need to be re-tuned once the domain of the data changes, thus limiting it's adaptability / ease-of-use. This paper does not present results on different domains (eg. outdoors as presented in [14]).

**Quality Of The Limitations Section:**

Limitations are addressed clearly

**Questions For Rebuttal:**

1) Please comment on the adaptability of this method to different domains (in terms of the many manual thresholds used).

2) line 207: \theta_{recall} is undefined.

**Robotics Focus:**

Sufficient demonstration on hardware

**Summary Of Paper:**

Developing on the work of OpenScene [14] which pioneered open-vocabulary 3D segmentation without training on 3D data, this work extends that idea by adding the capability of instance segmentation, thus allowing tasks such as robot manipulation. Despite the similarity in method formulation with [14], the resulting properties and performed tasks (real robot experiments) are sufficiently novel and present a useful contribution.

**Summary Of Recommendation:**

I believe this paper presents a useful research contribution and it's findings/proposed-method is likely to have high impact. I have noted a couple minor weaknesses but I am willing to recommend accepting this paper despite those (as the strengths far outweigh the weaknesses).

Post rebuttal update:
I will be keeping my original rating for the following reasons:
- I feel that the authors response to my review is adequate but not strong.
- I have taken the other reviews into account; which noted additional concerns regarding this work.

---

### Official Review · Reviewer_dbTB · 2023-07-18

**Confidence:** 4
**Originality:** Good
**Technical Quality:** Very Good
**Clarity Of Presentation:** Very Good
**Impact:** 3

**Recommendation:**

Weak Accept: I recommend accepting the paper, but will not argue for my recommendation if the majority of other reviewers have a different opinion.

**Review:**

This paper introduces a novel methodology for the semantic alignment and retrieval of 3D object instances, presenting an effective open-vocabulary approach that does not necessitate explicit training with 3D data

1. Quality

The paper is nicely written and structured. The problem well exposed and easy to understand. The method clearly exposed, and results are provided for the main standard tracking benchmarks. However, the 2D to 3D projection may miss some information.

2. Clarity

This paper is clearly written and easy to read.

3. Originality

This paper represents an incremental advancement in the field, distinguishing itself from existing works such as OpenScene through its direct projection of 2D results into the 3D space. By accomplishing the seamless integration of both 2D and 3D modalities, this paper pioneers a distinctive approach that relies on fundamentally distinct algorithms and methodologies. This originality arises from the novel fusion of 2D and 3D information, resulting in an innovative framework for semantic alignment and retrieval of 3D object instances

4. Significance

It has certain significance to introduce 2D algorithm into 3D task.

Strengths:
1. Straightforward but good pipeline to obtain 3D semantic scene representation.
2. Do not need any 3d labels.

Weakness:
1. 2D-3D projection is tricky? What if the depth information is noisy?
2. 3D filtering and association is also problematic for small objects and occluded objects

**Quality Of The Limitations Section:**

Limitations are not well addressed

**Questions For Rebuttal:**

Q1: Running time is missing.

Q2: If no ground truth depth information is available, how to conduct 2D-3D projection? Will it affect the quality of the overall pipeline?

Q3: How to deal with small objects and occluded objects in 3D filtering and association?

**Robotics Focus:**

Sufficient demonstration on hardware

**Summary Of Paper:**

This paper presents a novel approach for semantic alignment and retrieval of 3D object instances, termed as an effective open-vocabulary method that does not rely on explicit training with 3D data. The primary objective of this method is to provide a ranked set of 3D instance segments based on their semantic similarity to a given text query.

The proposed pipeline consists of several stages. A 2D Open-Vocabulary Instance Segmentation Method is pre-trained to extract text-aligned features and generate 2D region proposals. To enable accurate matching between the 2D regions and 3D instances, a two-stage object detection architecture is employed. The matching process relies on cosine similarity to establish the correspondence between 2D and 3D representations, followed by fusion of the matched instances in the 3D space. Furthermore, post-processing steps involving filtering and merging of the fused results are applied to enhance the overall accuracy and coherency.

**Summary Of Recommendation:**

Useful and straightforward pipeline towards 3D open vocubulary scene understanding. But some important questions need to be clarified.


Update: Thanks for the efforts of authors. After reading the feedback from the authors and going through the comments of other reviewers. The reviewer agree's that this work is a good “engineered” solution to achieve open vocab 3d segmentation, which is good to the community but not that novel. As a result, the reviewer would like to keep to the original scores.

---

### Official Review · Reviewer_MRDv · 2023-07-24

**Confidence:** 5
**Originality:** Good
**Technical Quality:** Fair
**Clarity Of Presentation:** Good
**Impact:** 3

**Recommendation:**

Weak Accept: I recommend accepting the paper, but will not argue for my recommendation if the majority of other reviewers have a different opinion.

**Review:**

## Strengths

The paper is well-written and easy to understand. The approach is simple, and explained in a concise manner, with sufficient detail to enable a skilled practitioner to implement it. From a sequence of input images, an open-vocabulary object detector (here, Detic) is used to obtain 2D bounding boxes. Each bounding box is tagged with a text label (one of 21k ImageNet categories), which is passed through CLIP to produce a text embedding for that label. Assuming depth images and camera poses to be known, the 2D proposals may be lifted to 3D using criteria such as bounding box overlap and CLIP feature similarity; and postprocessed to remove duplicate 2D (or 3D) detections. This results in a 3D map where each instance is additionally tagged with a CLIP embedding. Given a text query, the query is encoded using the CLIP text encoder, and compared with the stored text embeddings of each 3D instance via cosine similarity. The system then returns a ranked list of object instances that best match the text query.

This system is quite simple, yet performant. The system achieves better performance than OpenScene -- a recent method that computes a per 3D point CLIP descriptor (and consequently only performs semantic, not instance, segmentation) for open-vocabulary 3D detection. It also performs better than Fusion++ and PanopticFusion, which are object-centric mapping approaches that lift 2D object detections to 3D.

The qualitative results in the supplemental video demonstrate the crisp object instances and accurate retrievals across one small apartment and another tabletop scene.


## Weaknesses

There are a number of issues with the current edition of the manuscript that I would like to see discussed/addressed before I can recommend acceptance.

[W1] **Dependence on detection and description schemes**: The proposed method seems to be fairly agnostic to the exact open-vocabulary object detector and the feature descriptor used. Yet somehow, the current manuscript describes the method as being only applicable to DETIC detections and CLIP text feature descriptors; because of which the core contribution (i.e., a lightweight 3D lifting mechanism for 2D bounding box detections) does not seem to get its due credit (and also makes it very difficult to attribute credit to each aspect of the pipeline).

* The presented 3D lifting scheme is lightweight and performant, regardless of whether the object detector is closed- or open-vocabulary; and is agnostic to the choice of open-vocabulary detector. To establish this claim, one would need to evaluate the proposed method with a closed-vocabulary object detctor; and also with other open-vocabulary object detectors (popular open-source examples include Grounded-DINO, ViLD).

* The description scheme used in the proposed method is to take the Detic text label for each bounding box and encode the label using the CLIP text encoder. However, this seems to defeat the purpose of using image-text aligned embeddings (like CLIP) in the first place; since the image content is never used at subsequent stages in the pipeline. Only the text labels end up being used; and only text queries are submitted to the system at inference time. If only text queries are being used, one could alternatively try out embeddings like RoBERTa and GloVe, which will likely provide similar or better similarity measures, while being more lightweight. (on the other hand, if the usage of image-language aligned features is to be justified, the text CLIP embeddings will need to be replaced by the image-level CLIP embeddings).


[W2] **Evaluation on open-set categories**: Another key issue with the current evaluation scheme is that since the object detector Detic is trained using LVIS, CC, and ImageNet-21K, which collectively include most (if not all) of the ScanNet 200 categories. This effectively reduces the problem addressed in this paper to a closed-set / long-tailed detection problem setting, opposed to a true open-set recognition one. Apart from ScanNet-200, the only other dataset evaluated is YCB-Video, but there are only 12 videos in that dataset (the validation split). Evaluating on truly open-set scenarios is indeed a challenge, as pointed out by ConceptFusion [39] and LERF [40] (both cited here-in; and proposing to collect open-set data of their own). It is therefore important to evaluate the proposed method on open-set recognition scenarios (i.e., categories that the underlying object detection model has not seen during training time). While Table 3 attempts to evaluate "ImageNet21k - ScanNet200" categories, that settting too, is not truly open-set (because the underlying model used, Detic, has still been trained on the LVIS, CC, and ImageNet-21K datasets). One concrete example for an open-set scenario is another CoRL paper from last year "Semantic Abstraction" (https://semantic-abstraction.cs.columbia.edu/), which evaluates on the ARKitScenes dataset.

[W3] **Evaluation concerns with using only mAP**: Table 1 and Table 2 compare mAP scores across the proposed method, Fusion++, and PanopticFusion. Each of these methods reconstructs a different pointcloud or voxelgrid; owing to their customized noise-removal and filtering strategies. How is consistency ensured when comparing mAP metrics across these different pointclouds? (e.g., if the pointclouds from the proposed approaches end up conservatively removing noisy points; while the baselines do not; the mAP numbers for the proposed approach will likely end up being better) What I'm trying to get at, is that the evaluation protocol ends up conflating errors due to incorrect map construction with errors induced due to imperfect segmentation. This can be mitigated, in part, if reacalls and F1 scores were to be computed and reported.


# Post-rebuttal update

The author response addresses my concerns [W1] and [W3] adequately. In terms of [W2], however, a few concerns still remain, necessitating a more nuanced discussion. Since the author response only came in towards the very end of the discussion phase, this did not allow for reviewer-author interaction. That said, I appreciated the thoroughness and thought that went into the author response, and would like to revise my score to a "weak accept", but with certain caveats.

* The evaluation in the paper does not sufficiently demonstrate open-set properties, compared to the queries demonstrated in other recent relevant methods such as LERF, ConceptFusion, CLIP-Fields, and variants. Even in the additional experiments showcased in the rebuttal, *evaluation is restricted to synonyms of classes that appear in the ScanNet200 dataset*, limiting the "open-setness" claims of the system. I would strongly recommend adding a few results (qualitative results could be okay too) on scenes from NYUv2 or ARKitScenes, following SemanticAbstraction -- or demonstraing stronger open-set results using other appropriate datasets. The authors agree with part of my characterization of their open-set claims, and also say in the rebuttal that they do.

* I agree with the authors' characterization that the proposed approach will attempt to preserve DETIC's performance, and *lift it to 3D*.

* I would like this discussion/caveat on "open-set" evaluation to feature at a prominent location in the revised manuscript.

To summarize, I find [W1] and [W3] adequately addressed. [W2] is not completely addressed, but with the amends suggested in the author response, that to me tips the scales just about enough in favor of a weak accept rating.

**Quality Of The Limitations Section:**

Additional details required

**Questions For Rebuttal:**

I would like to see "Weaknesses" [W1], [W2], and [W3] addressed.

**Robotics Focus:**

Relevant but unlikely to deploy to hardware in near future

**Summary Of Paper:**

This paper presents an approach for open-vocabulary 3D object detection that is based on a straightforward lifting of 2D open-vocabulary object detections (bounding boxes) to 3D (assuming known depth and camera poses). The fused objects in 3D each have a CLIP embedding, which enables open-vocabulary (text-based) querying. The approach is evaluated on ScanNet-200 and YCB-Video.

**Summary Of Recommendation:**

While the paper presents an open-vocabulary 3D detection scheme of potential interest to the robot perception community, there still remain a number of design choices and evaluation-related concerns that need to be addressed before I can recommend acceptance. While I will, for now, score this a weak reject, I am very much open to revising my rating should my concerns be adequately addressed.

---

### Official Review · Reviewer_dozd · 2023-07-28

**Confidence:** 4
**Originality:** Good
**Technical Quality:** Very Good
**Clarity Of Presentation:** Very Good
**Impact:** 3

**Recommendation:**

Weak Accept: I recommend accepting the paper, but will not argue for my recommendation if the majority of other reviewers have a different opinion.

**Review:**

This paper is well written and mostly easy to understand. The paper is validated on standard datasets and with an extensive ablation studies to assess the benefits of each part of OVIR-3D. Despite these undeniable strengths, the reviewer considers OVIR-3D as an
"engineered" solution combining several existing approaches. Thus, the technical novelty and contribution of this paper are somehow limited.
On the other hand, in OVIR-3D, the different existing approaches are combined in a clever and efficient way. This paper might be of interest for CoRL community.
The paper should nonetheless be improved and clarified on certain aspects as detailed in the "Questions for Rebuttal" section.

**Quality Of The Limitations Section:**

Additional details required

**Questions For Rebuttal:**

1) The authors did not clearly motivate their work. Why is it useful to do 3D instance retrieval when there are a lot of existing 2D methods?

2) The proposed matching is unclear. The reviewer understand the cosine similarity but what about the region overlapping? How is the potential rotation  (depending on the point of view) of the 3D objects addressed since it seems that there is no registration?

3) It is not clear for the reviewer how to separate the 3D instances from the proposed approach. Let's say that there are 2 instances of the same objects in different areas of the environment. Their features may certainly match as well as their overlapping region.

4) The reviewer does not understand the 3D features  $f_i^{3D}$. Since the memory bank is initially empty, it can only be filled with $f_i^{2D}$. How does $f_i^{2D}$ become $f_i^{3D}$?

5) Please detailed, the different steps required to transfer to a real robot as shown in Section 7. From the reviewer understanding, on a fixed arm, there is no exploration (or is there a scanning phase?), thus less redundant region proposal. Is the proposed method robust enough to work with a single view?

6) Related to 5),  it would be interesting to quantify the robustness of the proposed approach given the number of views.

7) Minor typo: line 261: It's -> It is

**Robotics Focus:**

Sufficient demonstration on hardware

**Summary Of Paper:**

This paper presents OVIR-3D, a 3D instance retrieval from a text input. Given an explored 3D environment and a open-vocabulary query, the proposed method is able to rank all likely 3, without training on 3D data. The method uses an 2D object detector (Detic)  features that are first aligned with text features using CLIP.  The region obtained from Detic are then projected in 3D and compared (with cosine similarity features and overlapping) to existing 3D instances.

**Summary Of Recommendation:**

The paper is well written and clear. Although contribution is not technically novel, it is well-executed, with convincing results.
Nonetheless, there are still some technical points to clarify before recommending this paper for CoRL.

Update:
Given the clarification and additional evaluation provided by the authors, the reviewer recommends a Weak Accept for this paper.

---

### Decision · Program_Chairs · 2023-08-30

**Decision:**

Accept (Poster)

**Comment:**

### Summary, strengths and weaknesses
This paper introduces a method for open-vocabulary 3D object retrieval without 3D training. This approach combines 2D-to-3D lifting with CLIP embeddings for open-vocabulary querying, demonstrating a novel method for semantic alignment and retrieval of 3D object instances. Although the method is simple, the experimental results using standard datasets show that the method is performant.

The reviewers initially raised a few concerns regarding the method and the experimental comparison. The authors provided a detailed response to each of these points. The reviewers agree that most of the concerns have been addressed.

In their final scores, all the reviewers agree on the quality and acceptance. I agree with their consensus.